# Genomic regression analysis of coordinated expression

Ling Cai[1,2], Qiwei Li[2], Yi Du[3], Jonghyun Yun[4], Yang Xie[2], Ralph J. DeBerardinis[1] & Guanghua Xiao[2]

Co-expression analysis is widely used to predict gene function and to identify functionally related gene sets. However, co-expression analysis using human cancer transcriptomic data is confounded by somatic copy number alterations (SCNA), which produce co-expression signatures based on physical proximity rather than biological function. To better understand gene–gene co-expression based on biological regulation but not SCNA, we describe a method termed "Genomic Regression Analysis of Coordinated Expression" (GRACE) to adjust for the effect of SCNA in co-expression analysis. The results from analyses of TCGA, CCLE, and NCI60 data sets show that GRACE can improve our understanding of how a transcriptional network is re-wired in cancer. A user-friendly web database populated with data sets from The Cancer Genome Atlas (TCGA) is provided to allow customized query.

[1] Children's Medical Center Research Institute at UT Southwestern Medical Center, 6000 Harry Hines Blvd, Dallas, TX 75235, USA. [2] Quantitative Biomedical Research Center at UT Southwestern Medical Center, 5323 Harry Hines Blvd, Dallas, TX 75390, USA. [3] Department of Bioinformatics at UT Southwestern Medical Center, 5323 Harry Hines Blvd, Dallas, TX 75390, USA. [4] Department of Mathematics at University of Texas at Arlington, 411S. Nedderman Drive, 478 Pickard Hall, Arlington, TX 76019, USA. Correspondence and requests for materials should be addressed to R.J.D. (email: Ralph.Deberardinis@UTSouthwestern.edu) or to G.X. (email: Guanghua.Xiao@UTSouthwestern.edu)

Highly coordinated expression of genes functioning in common processes is a widespread phenomenon in prokaryotes and eukaryotes[1, 2]. Spatial and temporal compartmentalization of gene expression in response to environmental cues allows cells to avoid futile reactions in metabolism, promotes efficiency in the stoichiometric assembly of macromolecular complexes, and reduces noise in signal transduction pathways, among other functions. Co-expression analysis is a widely adopted tool for functional prediction and identification of functionally related gene sets. Such analysis identifies associated genes based on their highly correlated expression profiles from high-throughput gene expression profiling data obtained by microarray or RNA-seq experiments[3].

Great efforts from multiple projects such as The Cancer Genome Atlas (TCGA) and Cancer Cell Line Encyclopedia (CCLE)[4, 5] have been put forth to characterize cancer genomics from tumor tissues and cell lines. A rich resource of cancer transcriptome data has been generated from these studies. Many popular cancer data-mining platforms such as Oncomine, cBio-Portal, and CCLE provide co-expression analysis tools based on these cancer transcriptome data[4, 6, 7]. With these tools, users can search for genes co-expressed with a gene of interest in different tumor types or in cancer cell lines. However, results from co-expression analyses using standard methods are often hard to interpret because many top correlating genes arise from somatic copy number alteration (SCNA) rather than bona fide transcriptional regulation. Copy number alteration (CNA) is a common feature in cancer[8]. Multiple studies have shown that gene expression levels are correlated with copy numbers in cancer cells[9, 10] as well as cultured human pluripotent stem cells[11]. Notably, Fehrmann et al., who re-analyzed 77,840 expression profiles, found that 99% of all abundantly expressed human genes exhibited a positive correlation with DNA copy number[12]. Dosage-compensation mechanisms, which ensure that genes are expressed at appropriate levels irrespective of their copy numbers, have not been found to exist for autosomes in humans[13]. Hence, variation of RNA levels in cancer samples is a combined consequence of CNA and biological regulation of transcript production, processing and decay. Several groups have used copy number-adjusted expression values to understand transcriptional consequences of genomic aberrations[14, 15]. However, no method exists to remove the confounding effect of CNAs in the analysis of gene–gene co-expression using cancer transcriptome data.

To better understand gene–gene co-expression based on biological regulation but not SCNA, here we describe a method named genomic regression analysis of coordinated expression (GRACE) to adjust for the effect of CNA from co-expression analysis. Through comprehensive analyses of genetics, genomics, proteomics, metabolomics, and drug response data from the public domain, we show that GRACE can improve our understanding of how a transcriptional network is re-wired in cancer. A user-friendly web database has also been built with data from multiple TCGA cohorts to allow for customized query.

## Results

### SCNA in cancer cause co-expression of neighboring genes.
SCNA is frequently observed in cancer[8]. In normal tissues where all cells are diploid, gene–gene correlations are usually a result of coordinated biological regulation to allow concerted expression of genes with related functions. In the presence of SCNA, the expression of genes becomes proportional to the copy number, while copy number changes in functionally related genes are often independent of each other. This results in a reduction in correlation between functionally related genes and an increase in correlation between neighboring genes situated in the same DNA

segment that is affected by the same CNA event (Fig. 1a). The impact of SCNA confounds the interpretation of co-expression genes (Fig. 1b). Here we provide some examples based on over one thousand tumor samples from TCGA Breast Invasive Carcinoma (BRCA) cohort. Significant focal and arm level SCNA events have been found for these samples[16]. For example, chromosome 1p is frequently deleted, while chromosome 1q is frequently amplified (Fig. 1c). We examined the impact of SCNA on mRNA abundance in this cohort. A correlation heatmap revealed a strong positive correlation between expression levels and copy number values for genes in the same arm of chromosome 1 (Fig. 1d), and a similar pattern was also observed in other chromosomes. Importantly, because SCNA occurs in segments that cover multiple genes, RNA levels of a gene not only correlate with its own copy number values but also broadly correlate with the copy numbers of the physically neighboring genes, which causes the co-expression of neighboring genes (Fig. 1b, e). This bias toward positive correlation between RNA and copy number occurs for genes within the same chromosome but not for genes from different chromosomes (Fig. 1f)

### GRACE adjusts for effects of SCNA in co-expression analysis.
To adjust for the variation in gene expression contributed by SCNA, we fit a linear regression model using the copy number values as the predictor variable and RNA levels as the response variable. The residuals from this linear regression model represent variations in gene expression that could not be accounted for by SCNA. To apply this method, samples with both gene expression and copy number data were selected. Genes were filtered to remove under-expressed genes or genes with saturated copy number values so that both the RNA and copy number of the same gene could have an approximately linear relationship ("Methods"). The residuals for each gene were then calculated and used for subsequent co-expression analysis (Fig. 2a). Collectively, we refer to this method as genomic regression analysis of co-expression (GRACE). We use an autocorrelation plot to visualize the effect of our method in removing correlation from neighboring genes due to SCNA. Autocorrelation was calculated for all genes and averaged over all tumor samples or all normal samples. The average autocorrelation of gene expression in tumor samples is much higher than that of the normal samples presumably due to SCNA, and adjusting expression with copy number data brings down the gene–gene autocorrelation in tumor (Fig. 2b).

In Fig. 2c–h, we provide a specific example using *EIF2D*, a gene encoding eukaryotic translation initiation factor 2D[17], to compare the results from co-expression analysis with the standard method or GRACE using data from the TCGA BRCA cohort.

*EIF2D* is located on chromosome 1q32.1. In TCGA BRCA tumor samples, the gene expression levels of *EIF2D* positively correlate with its copy number levels (Fig. 2a) as a result of the frequent amplification events of chromosome 1q (Fig. 1a). The residuals from regressing gene expression levels of *EIF2D* on the copy number levels of *EIF2D* were calculated to represent the variations in *EIF2D* transcript levels after adjusting for its copy number levels (Fig. 2a). The standard method of co-expression analysis calculates expression correlation among genes based on gene expression levels and sorts the genes by strength of positive correlation, and the resulting top 10 *EIF2D* co-expressing genes are all from chromosome 1q, neighboring the *EIF2D* locus (Fig. 2c). In contrast, using our method GRACE, which calculates correlation based on copy number-adjusted gene expression levels, the resulting top 10 *EIF2D* co-expressing genes are from various chromosomes and are almost exclusively ribosomal protein genes involved in translational processes similar to *EIF2D* (Fig. 2d). An extended view of the top 200 *EIF2D* co-expressing

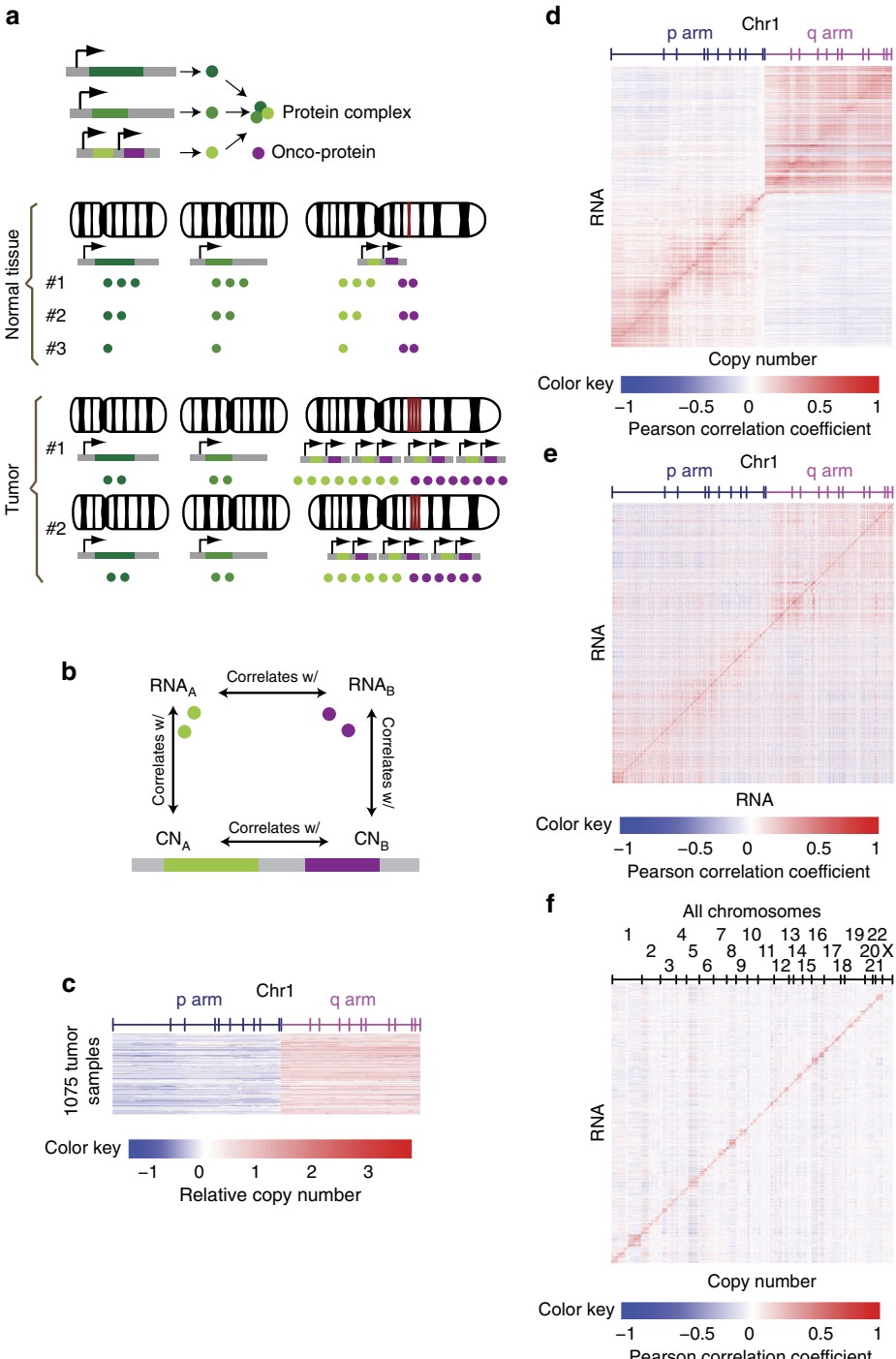

**Fig. 1** Increased correlations between genes from neighboring loci as a result of copy number variation. **a** Schematic diagram illustrating an example of copy number alteration confounding co-expression analysis. In this example, three subunits of a protein complex are encoded by three genes from different chromosomes. In normal tissues, coordinated transcriptional regulation ensures equal amounts of three subunits are produced to facilitate stoichiometric assembly of the complex. Transcript levels of these three genes will therefore be highly correlated. In tumor samples, the third gene is amplified together with a neighboring oncogene with unrelated functions. Consequently, correlation of the third gene with the other two related genes decreases, whereas its correlation with the unrelated oncogene neighbor increases. **b** Schematic diagram delineating correlation relationships between copy numbers of neighboring genes, between RNA levels of neighboring genes and between copy number and RNA levels of the same gene as a result of CNA events in cancer. **c** Relative copy number for genes on Chromosome 1 in 1075 tumor samples from TCGA breast cancer cohort. Genes are ordered by location on chromosome, and borders of cytobands from p and q arms are marked on top. P arm is frequently deleted and q arm is frequently amplified. **d** Copy number-RNA Pearson correlation matrix for genes from chromosome 1. Arm-level copy-number alteration events from chromosome 1 results in positive correlation between copy number and RNA for genes from the same arm. **e** RNA–RNA Pearson correlation matrix for genes from chromosome 1. Co-amplification or co-deletion of genes located in the same segment result in increased positive correlation between neighboring genes. **f** Copy number-RNA Pearson correlation matrix for all genes. Genes are ordered by location in chromosomes, and names of chromosomes are given on top. Increased positive correlation is observed only for genes from the same chromosome or chromosomal arm but not for genes from different chromosomes

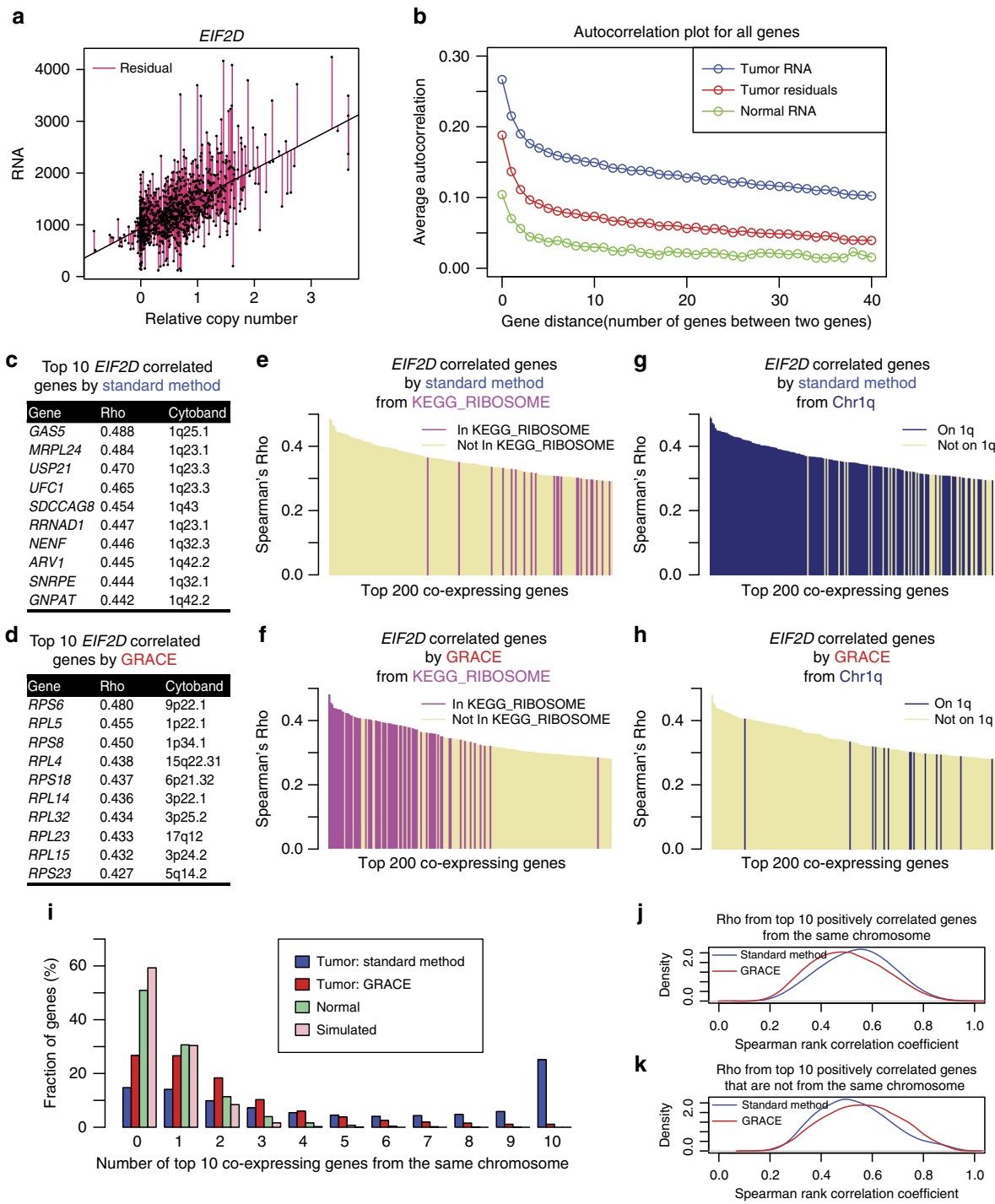

**Fig. 2** Genomic regression analysis of co-expression (GRACE) corrects for correlation bias due to copy number variation. **a** Relative copy number levels vs. RNA levels of *EIF2D* for all 1075 tumor samples. The residuals from the linear regression model fitted with all 1075 tumor samples were marked by purple lines. **b** Average autocorrelation of neighboring genes in TCGA BRCA samples. Correlations between neighboring genes separated by 0–40 genes in between were calculated as autocorrelation for each sample. The average of autocorrelations was taken for 1075 tumor samples and 112 normal samples. The autocorrelation of neighboring genes based on copy number-adjusted expression levels variation (tumor residuals) was markedly reduced compared to autocorrelation based on tumor RNA. Average autocorrelation from normal samples represent the gene–gene autocorrelation baseline in diploid cells. **c**, **d** Top 10 *EIF2D* correlated genes by standard method (**c**) or GRACE (**d**) based on Spearman correlation. Top 10 *EIF2D* correlated genes by standard method are all located in cytoband 1q whereas top 10 *EIF2D* correlated genes by GRACE are all outside cytoband 1q and are all ribosomal protein genes. **e**, **f** Enrichment of KEGG_RIBOSOME genes from the top 100 *EIF2D* correlated genes by standard method with a *p* value of 0.10 (**e**) or by GRACE with a *p* value of 9.0e−143 (**f**). **g**, **h** Enrichment of genes from chromosome 1q in the top 100 *EIF2D* correlated genes by standard method with a *p* value of 8.7e−119 (**g**) or by GRACE with a *p* value of 0.99 (**h**). **i** Relative frequency distribution of chromosomal neighbors in top 10 co-expressing genes for all genes. GRACE markedly reduced the number of chromosomal neighbors in the top 10 co-expressing genes. **j**, **k** Kernel density estimation plots that visualize the distribution of pooled Spearman rank correlation coefficients for top 10 co-expressing genes from the same chromosome (**j**) or not from the same chromosome (**k**). Compared to the standard method, GRACE decreased intra-chromosomal gene–gene correlation and increased inter-chromosomal gene–gene correlation. All analyses are based on TCGA BRCA data

genes by the standard method or by GRACE again shows that genes identified by the standard method are heavily biased toward genes in the physical proximity of *EIF2D*, while genes identified by GRACE bear more biological relevance to the function of *EIF2D* in protein translation (Fig. 2e,g,f,h).

Following this specific example, we also performed a systematic comparison between GRACE and the standard methods using the TCGA BRCA data. For every gene, we counted how many of its top 10 co-expressing genes were from the same chromosome on which this gene is located. This count of chromosomal neighbor genes ranges from 0 to 10. The relative frequency distributions of these counts are given in Fig. 2i. With the standard method, over 25% of genes have all of the top 10 co-expressing genes coming from the same chromosome, while this number is reduced to <2% with GRACE. GRACE has largely reduced the number of physically neighboring genes in the top 10 co-expressing genes. We have also calculated such distribution using normal samples from the TCGA BRCA cohort, which are presumably all diploid cells. Furthermore, we computed the simulated distribution assuming all genes are randomly distributed across 23 chromosomes (the Y chromosome is excluded from the TCGA BRCA data). The normal samples have more co-expressing genes from the same chromosome than the simulated distribution, which agrees with the notion that gene order in the human genome is not completely random and certain clusters of genes with functional relevance are co-expressed[18].

Next, we compared the strength of correlation among genes from the same chromosome and among genes from different chromosomes. As expected, the overall correlation among genes from the same chromosome is reduced with GRACE compared to the standard method (Fig. 2j). On the other hand, correlations among genes from different chromosomes are improved with GRACE compared to the standard method (Fig. 2k). This increase in significance of inter-chromosomal gene–gene co-expression by GRACE is likely a result of SCNA noise removal in co-expression analysis and could help us better understand the biological regulation of transcript levels in the cancer context. Similar results are also observed with the CCLE cell line data and METABRIC discovery set data[4, 19] (Supplementary Fig. 1).

**GRACE facilitates discovery of tumor-specific co-expression**. Tissue-specific gene networks are useful in capturing tissue-specific functional interactions[20]. But in cancer, the transcription network could be rewired to support cancer-specific needs. Here with two specific examples, we show that GRACE can help us better understand how coordinated gene expression is rewired from normal to tumor tissues. Supplementary Data 1 include the top results from systematic identification of tumor-unique- or normal-unique-coexpressing gene enrichment in gene families from HUGO Gene Nomenclature Committee (HGNC) classification[21] or canonical pathway gene sets from the Molecular Signatures Database (MSigDB) collection[22].

In the first example, we examined the difference in co-expressing genes for Poly (ADP-ribose) polymerase-2 (*PARP2*) in tumor tissues vs. normal tissues (Fig. 3a–c). *PARP2* is a member of the PARP family. The founding member *PARP1* is a cancer therapeutic target famous for its role in detection of DNA damage and recruitment of various proteins for DNA repair[23]. PARP inhibitors that target both *PARP1* and *PARP2* have been used to treat *BRCA1*- and *BRCA2*-deficient tumors and the synergism in conjunction with DNA damaging agents is also being tested in various clinical trials[23]. Despite their similar role in DNA repair, *PARP2* lacks the N-terminal DNA-binding zinc fingers in *PARP1*[24]. With the TCGA BRCA tumor samples, the top *PARP2* co-expressing genes found by the standard method

are mostly genes located near the *PARP2* locus at chromosome 14q11.2, whereas the top co-expressing genes called by GRACE are from various chromosomes and many are involved in cell cycle processes (Fig. 3a). The top *PARP2* co-expressing genes found using the normal samples, however, are far less related to cell cycle. Only two of the top 100 *PARP2* co-expressing genes in BRCA normal samples belong to the "REACTOME_CELL_-CYCLE" gene set, while close to half of the top 100 *PARP2* co-expressing genes in BRCA tumor samples called by GRACE belong to this gene set (Fig. 3b). This difference between *PARP2* co-expressing genes in tumor samples and in normal samples is consistently observed over multiple TCGA cohorts (Fig. 3b). Moreover, genes belonging to the C2H2-type zinc fingers gene family from the HGNC database[21] are frequently found in the top 100 *PARP2* co-expressing genes from normal tissues of several TCGA cohorts (Fig. 3c). This difference in the function of *PARP2* co-expressing genes between tumor samples and normal samples suggests that *PARP2* might be highly involved in DNA damaging repairs in tumor because of the genomic instability in cancer, while in normal tissues it might be involved in processes that also require C2H2-type zinc finger proteins, such as transcriptional regulation[25]. The levels of *PARP2* transcripts are similar in normal and tumor samples, whereas *PARP1* has on average a twofold increase in tumor samples compared to normal samples (Supplementary Fig. 2a, b). In addition, *PARP1* co-expressing genes from normal tissues are not enriched in C2H2-type zinc finger genes, suggesting that unlike *PARP2*, *PARP1* is not functionally related with the C2H2-type zinc finger transcription factor (Supplementary Fig. 2d).

In the second example, we examined the co-expressing genes for *CCT4*. *CCT4* encodes for a subunit of the chaperonin-containing TCP1 complex (CCT). The CCT complex assists the folding of newly translated peptides, including actin and tubulin[26]. As expected, *CCT4* co-expresses with genes encoding other subunits of the CCT complex in tumor, and this is better detected by GRACE than by the standard method (Fig. 3d, e). Interestingly, *CCT4* has a much tighter correlation with ribosomal genes in normal samples in several TCGA cohorts (Fig. 3d–f). While the co-expression of *CCT4* and ribosomal proteins might suggest better coupling of peptide folding and translation processes in normal tissue, it is also possible that monomeric *CCT4* could engage in functions independently of the CCT oligomer[27].

**Applying GRACE in multi-dimensional data analysis**. Cancer cell lines are widely used as in vitro models for biological research. Many cancer cell lines have been extensively profiled for their copy number, transcriptome, proteome, metabolome and drug sensitivity, among others. Because SCNA also cause a distortion of RNA levels in these cancer cell lines from what would normally occur for a diploid cell in the same biological context, we exploited the use of residuals (RES) from regressing RNA expression levels on copy number values to identify genes with copy number-adjusted RNA expression correlated with orthogonal molecular features. We then compared them with the standard approach using unadjusted RNA expression.

In the first example, we looked at the top 100 genes with RES or RNA levels correlated with the protein levels of Myc measured by reverse phase protein array (RPPA) in NCI-60 cell lines. Myc is a frequently amplified oncogene located on chromosome 8q. One of its critical roles in tumorigenesis is to broadly regulate transcription, in particular the ribosome biogenesis program[28]. We found a higher prevalence of genes belonging to a previously characterized Myc target gene set[29] or a ribosome gene set from the top100 Myc RPPA correlated genes by RES than by RNA

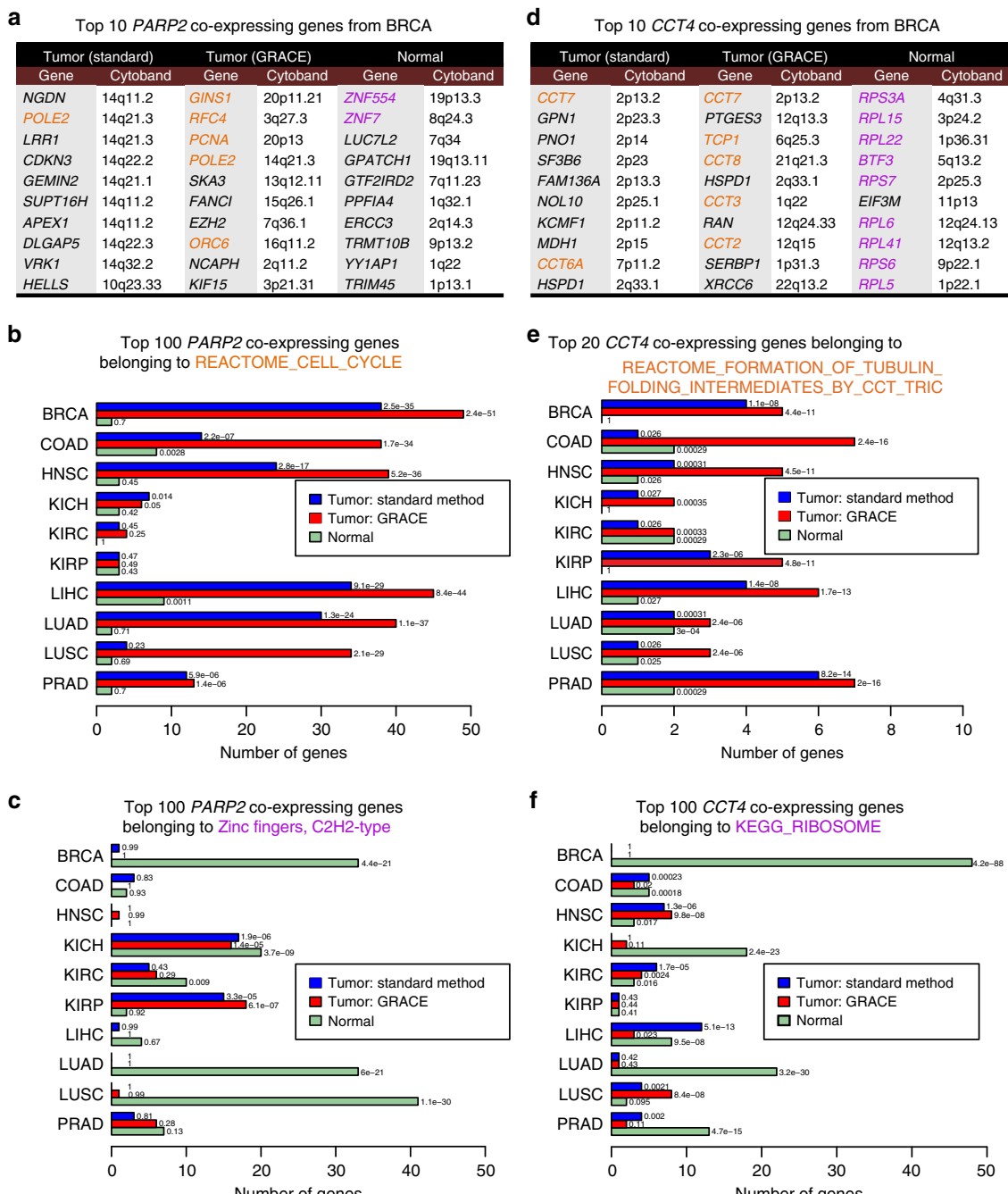

**Fig. 3** Examples of co-expressing genes differing in tumor vs. normal tissues. **a** Top 10 *PARP2* co-expressing genes from tumor (by standard method or GRACE) or normal tissue samples. Top *PARP2* co-expressing genes by standard method in tumors are dominated by genes from 14q, whereas top *PARP2* co-expressing genes by GRACE in tumors contain many known cell cycle genes (genes marked in orange are from gene set "REACTOME_CELL_CYCLE") pertinent to the known function of *PARP2* in DNA repair. Top *PARP2* co-expressing genes from normal tissues, however, do not contain cell cycle genes (genes marked in magenta are from gene family "Zinc fingers, C2H2-type"). **b, c** Frequencies of genes from "REACTOME_CELL_CYCLE" (**b**) or "Zinc fingers, C2H2-type" (**c**) in top 100 *PARP2* co-expressing genes in tumor (by standard method or GRACE) or normal tissues from different TCGA cohorts. *P* values from hypergeometric tests were given next to the bars. In many TCGA cohorts, cell cycle genes are highly enriched in *PARP2* co-expressing genes, and such enrichment is more significant with genes identified by GRACE than the standard method. In contrast, few cell cycle-related genes were found as *PARP2* co-expressing genes in normal tissues; instead, zinc finger proteins are highly enriched. **d** Top 10 *CCT4* co-expressing genes. Standard method found many *CCT4* physical neighbors on chromosome 2p as co-expressing genes whereas GRACE found many other CCT subunits that are in the same complex with *CCT4* (genes marked in orange belong to gene set "REACTOME_FORMATION_OF_TUBULIN_ FOLDING_INTERMEDIATES_BY_CCT_TRIC"). Top *CCT4* co-expressing genes from normal tissues, however, do not contain CCT subunits. Instead, many ribosomal protein genes were found (genes marked in magenta belong to gene set "KEGG_RIBOSOME") **e, f** Frequencies of genes from "REACTOME_FORMATION_OF_TUBULIN_ FOLDING_INTERMEDIATES_BY_CCT_TRIC" in top 20 *CCT4* co-expressing genes (**e**) or "KEGG_RIBOSOME" (**f**) in top 100 *CCT4* co-expressing genes. Analyses are based on TCGA BRCA data. *P* values from hypergeometric tests were given next to the bars. Abbreviations for TCGA cohorts: BRCA breast invasive carcinoma, COAD colon adenocarcinoma, HNSC head and neck squamous cell carcinoma, KICH kidney chromophobe, KIRC kidney renal clear cell carcinoma, KIRP kidney renal clear cell carcinoma, LIHC liver hepatocellular carcinoma, LUAD lung adenocarcinoma, LUSC lung squamous cell carcinoma, PRAD prostate adenocarcinoma

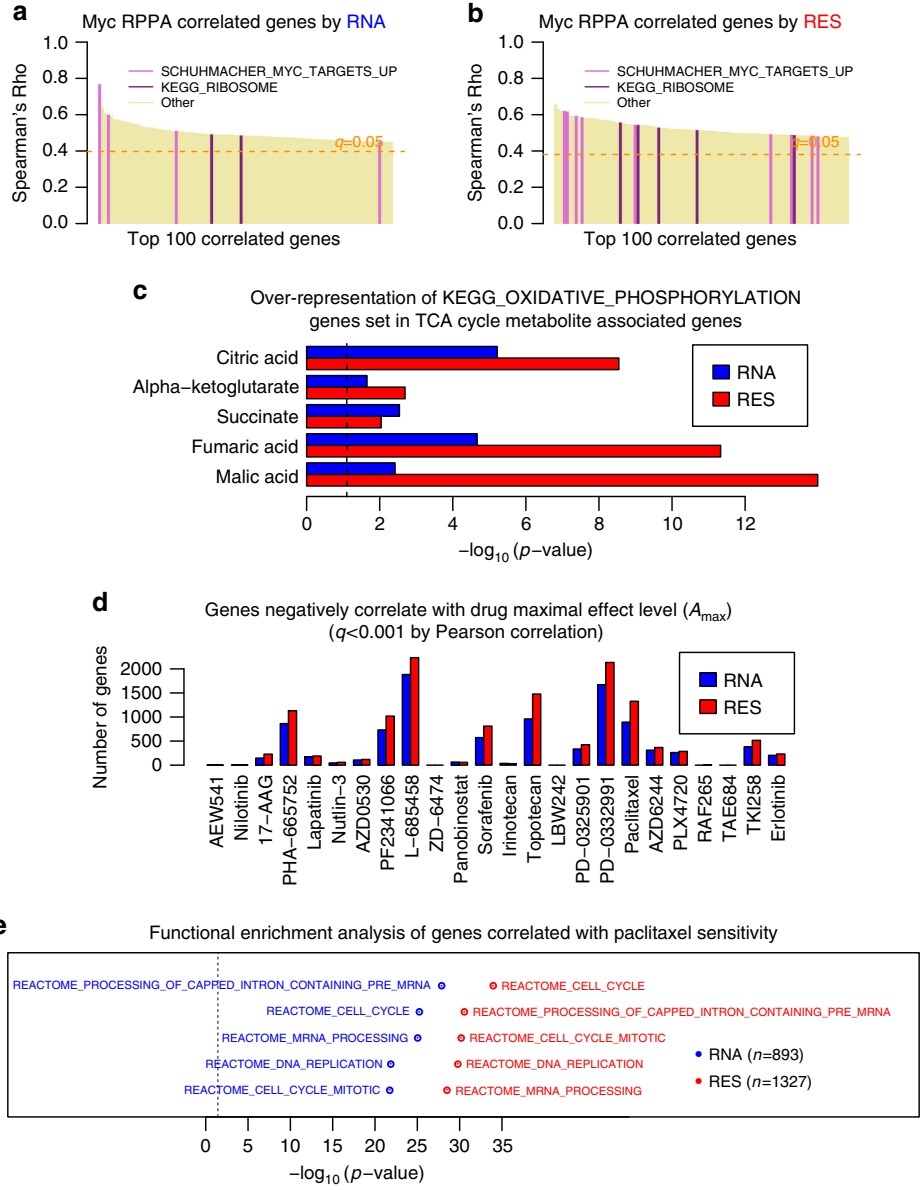

**Fig. 4** Correlation between transcriptomic data and orthogonal molecular features. **a**, **b** Enrichment of Myc transcriptional target genes from gene sets "SCHUHMACHER_MYC_TARGETS_UP" and "KEGG_RIBOSOME" in the top 100 Myc RPPA correlated genes based on RNA ($p$ value of 5.1e−4) (**a**) or RES ($p$ value of 1.2e−12) (**b**). Spearman correlation coefficient at false discovery rate of 0.05 based on the Benjamini & Hochberg method was marked by an orange dashed line. **c** Enrichment of "KEGG_OXIDATIVE_PHOSPHORYLTION" gene set in genes significantly (adjusted $p < 0.05$) correlated with multiple TCA cycle metabolite levels by RNA or RES. **d** Number of genes found to associate with drug sensitivity based on significant (adjusted $p < 0.001$) negative Pearson correlation between their expression levels and the high-concentration effect level (Amax) of the drugs tested. **e** Top 5 gene sets highly enriched in paclitaxel sensitivity correlated genes by RNA or RES. These include cell cycle and DNA replication gene sets, consistent with the role of paclitaxel in targeting mitosis. **a**–**c** Based on NCI60 cell line data, **d**, **e** are based on CCLE cell line data

(Fig. 4a, b). This result suggests that using copy number-adjusted RNA expression improves the correlation between transcription factor protein abundance and its transcriptional target levels (Fig. 4a, b).

In the second example, we used the metabolomics data of NCI-60 cell lines and compared genes correlated with the levels of several tricarboxylic acid cycle (TCA cycle) intermediates based on adjusted or unadjusted RNA expression data (Fig. 4c). Since the pool size of these metabolites could reflect activity of the TCA cycle that produces reducing equivalents that generate ATPs through oxidative phosphorylation, we tested the enrichment of genes involved in oxidative phosphorylation in the genes

significantly correlated with TCA cycle intermediates. While the results are all significant, for most metabolites, correlation based on RES outperformed RNA (Fig. 4c), suggesting the association between the metabolic status of the cell and the transcriptional program of oxidative phosphorylation is better detected when the noise from SCNA is reduced.

Lastly, we correlated gene expression to drug sensitivity using data from CCLE[4], using RNA or RES. Genes with negative correlation to the maximal effect level of each drug were selected based on a cut-off of adjusted $p$ value less than 0.001. Overall, correlation with RES identified more significant genes than with RNA (Fig. 4d), and this often resulted in more significant results

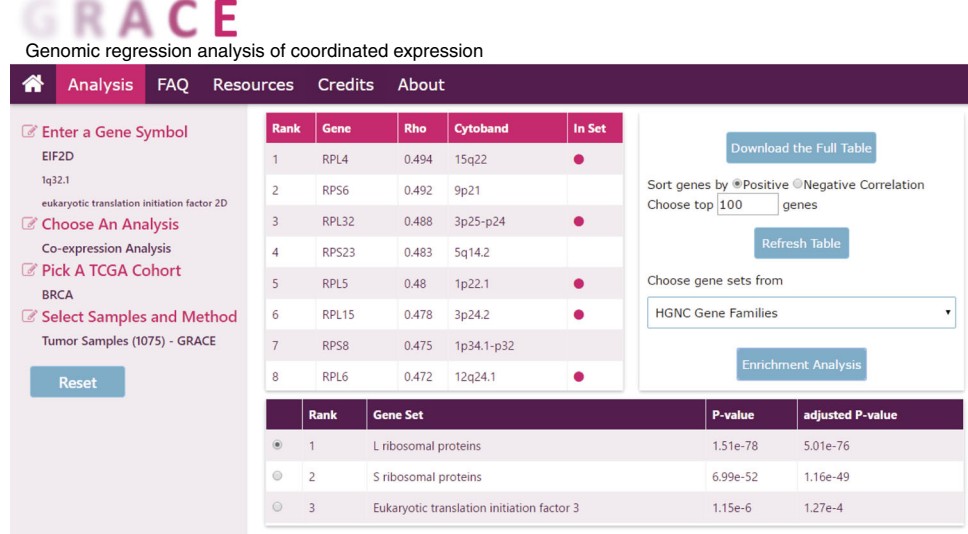

**Fig. 5** Co-expression analysis in web database GRACE. In the analysis page of the web database GRACE, upon entry of gene name, selection of "co-expression analysis" as the type of analysis and specification of cohort, sample types and method, a table of co-expressing genes will be generated. The result will include Spearman rank correlation coefficient and the chromosomal location for the co-expressing genes. Users may click the co-expressing gene names to be redirected to external websites for more information. Additional follow up analysis options are also provided. The user may download the full list of co-expressing genes, refresh the table with user specified sorting order and length of gene list, or run an enrichment analysis with a gene set library of their choice. In the resulting table of top enriched gene sets, the user may select from different gene sets and identify their members by the magenta dot in the "In Set" column of the co-expressing gene table

from functional enrichment analysis. For instance, paclitaxel, a drug that targets tubulin to interfere with mitotic spindle assembly, had more significant association with cell cycle-related gene sets based on correlation by RES compared to RNA (Fig. 4e), supporting the notion that cells actively engaged in proliferation could be more susceptible to drugs blocking cell division. This suggests that the transcriptional profile associated with drug sensitivity phenotype is better revealed without influence from SCNA.

For all three sets of orthogonal molecular data, we found that the number of significantly associated genes detect by the RES method was significantly larger than by the RNA method (Supplementary Data 2). One-sided $p$ values from Wilcoxon signed rank test are 7e−4, 4e−9, and 1e−4 for RPPA, metabolome, and drug sensitivity data respectively).

**The GRACE web database**. To introduce our method to the cancer research community, we have constructed a user-friendly open-access web database named GRACE (https://grace.biohpc. swmed.edu) using data sets from TCGA that cover over 20 disease cohorts. GRACE provides two analysis options, "RNA Copy Number Scatter Plot" and "Co-expression Analysis". In "RNA Copy Number Scatter Plot", users can visualize the relationship between the RNA expression and copy number for a gene of interest in tumor samples of a specific TCGA disease cohort (Supplementary Fig. 4). In "Co-expression Analysis", users can identify co-expressing genes for a specific gene in tumor or normal samples from a specific TCGA disease cohort. The standard method and GRACE are both provided for co-expression analysis of tumor samples. In addition, we also provide follow-up functional enrichment analysis of the co-expressing genes. Users may choose from different gene set databases to run the customized enrichment analysis. From the resulting enriched gene set table, users can select different gene sets to view the localization of genes from the chosen gene set in the co-expressing gene table (Fig. 5).

## Discussion

Amplification of oncogenes and deletion of tumor suppressor genes commonly take place in the process of tumorigenesis[8]. However, SCNA are often large-scale events that broadly affect many passenger genes besides the driver genes and thus negatively impact the fitness of the cells with increased replication stress, greater burden on the protein quality-control system, altered metabolism, etc.[13]. Measurement of copy number changes in cancer are critical in deciphering the driver events in cancer and explaining the disease phenotypes; in addition, passenger events from SCNA could also be exploited as specific vulnerabilities that could represent alternative therapeutic targets for cancer[30]. But from a standpoint of understanding transcriptional regulation in cancer, SCNA could be a big source of noise. Our work has revealed that teasing apart the contribution of SCNA from transcriptome data could improve the biological interpretation of co-expression analysis as well as the correlation between gene expression data and orthogonal molecular features. The benefits may be more obvious for genes severely affected by CNAs and for genes that are more tightly transcriptionally co-regulated with other genes. Our method can also help to discern the differences in transcriptional co-regulation between tumor and normal tissues. For example, we showed that whereas both *PARP1* and *PARP2* co-express with cell cycle genes in tumor samples consistent with their known roles in DNA repair, only *PARP2* strongly co-expresses with many C2H2-type zinc finger genes in normal tissues. It is possible that while the inhibition of *PARP1* by PARP inhibitors may provide therapeutic efficacy in cancer, the simultaneous inhibition of *PARP2* may result in toxicity via transcriptional dysregulation in normal tissues.

Our method does have several limitations. First, residuals are calculated by regressing RNA expression levels of a gene on its copy number levels. This simultaneous use of two types of data requires the exclusion of samples or genes available to only one type. Second, both data sets need to be in linear range in order for the linear modeling to be accurate. In some cases like the TCGA

BRCA copy number data, we found that copy number measurements were saturated for some highly amplified genes such as *ERBB2*, a gene encoding a receptor tyrosine kinase that is highly amplified in breast cancer (Supplementary Fig. 3a). These important genes unfortunately need to be filtered out. Genes with many undetected values also need to be excluded, and this is more common in microarray data, perhaps due to the lower sensitivity compared to RNA-seq.

We also noticed that GRACE did not fully correct the bias toward calling physical neighbors in the co-expression analysis (Fig. 2b, i, Supplementary Fig. 1a, d). This could be due to an over-simplified linear model. In our current study, we use pre-processed data from previous publications. It would be interesting to assess the impact of different data acquisition and processing procedures on this method.

Finally, the dynamics of gene–gene correlation or anti-correlation are much more significant in normal samples compared to tumor samples (Supplementary Fig. 3b, c). Although our method corrects for noise introduced by SCNA, there is little change to the strength of correlation (Supplementary Fig. 3b). Many factors could contribute to the perturbation of transcript homeostasis in cancer, such as genetic lesions, epigenetic changes, CNAs that result in aberrant levels of transcription factors, lncRNA, microRNA, RNA binding proteins, etc. Intratumoral heterogeneity, presence of stromal cells and infiltrating lymphocyte in impure bulk tumors can also add another layer of complexity to the tumor transcriptional profile (Supplementary Fig. 3d). There are still many challenges in untangling the dysregulated transcriptome in cancer and understanding the origin of cancer-specific transcriptional changes.

## Methods

**Gene co-expression analysis adjusted for copy number value**. In this study, we propose a practical but powerful method to adjust the effect of copy number on gene expression data in cancer samples. Let $Y_j$ and $X_j$ denote random variables for the gene expression and the copy number variation (CNV) of gene $j$, respectively. Without loss of generality, we assume these variables have zero mean and unit variance. We consider the following regression models describing the relation between two genes $j$ and $h$:

$$Y_j = \beta_j X_j + \alpha_{jh} Y_h + \varepsilon_j$$
$$Y_h = \beta_h X_h + \alpha_{hj} Y_j + \varepsilon_h,$$

where $\varepsilon_j$ and $\varepsilon_h$ are Gaussian noise with variance $\sigma_j^2$ and $\sigma_h^2$, respectively. The regression coefficient $\alpha_{jh}$ describes the association of gene expression between genes $j$ and $h$, adjusting for the CNV effect of gene $j$ and $h$. Our goal is to detect genes with high absolute values of $\alpha_{jh}$ and $\alpha_{hj}$ to identify co-expressed genes. However, the identification based on these two coefficients is computationally intensive, as it requires inference on the above regression models for all pairs of genes simultaneously. Instead, our proposed method uses the residual from a simpler regression model ($Y_j \sim \beta_j X_j$), and correlations between each pair of residuals are used to detect pairs of genes with high values of $|\alpha_{jh}|$ and $|\alpha_{hj}|$. We did simulation studies to compare the calculated coefficients using our approximation method with the true correlation coefficients conditional on the expression of all other genes. The simulation results show that our approach provides a good approximation of the true correlation values under most scenarios (Please see the simulation studies in Supplementary Note.). This approximation greatly improves the computation efficiency.

To calculate the correlation of the residuals, we regress the gene expression levels on the copy number values for each gene across different samples and take the residuals for subsequent co-expression analysis. Specifically, let $y_{ij}$ denote the gene expression of gene $j$ in patient $i$, and $x_{ij}$ denote the corresponding copy number value. For each gene $j$, we fit a simple linear regression model: $y_{ij} = b_{0j} + b_{1j} x_{ij} + e_{ij}$, and determine the fitted values $\hat{b}_{0j}$ and $\hat{b}_{1j}$ for the gene-specific coefficients $b_{0j}$ and $b_{1j}$. Finally, the residual $r_{ij} = y_{ij} - \hat{b}_{0j} - \hat{b}_{1j} x_{ij}$ is calculated and used as the copy number-adjusted gene expression value (for gene $j$ and patient $i$), as seen in the following co-expression analysis.

For standard co-expression analysis using the TCGA data, under-expressed genes with 0 values in over 10% of the total samples were filtered out. For GRACE using the TCGA data, we noticed the copy numbers for some highly amplified genes had saturated detection in the GISTIC copy number data. Hence, in addition to removal of the under-expressed genes same as above, we also removed genes with saturated copy number values in over 5% of the total samples.

Functional enrichment analysis was performed based on the hypergeometric test. Multiple comparisons correction was performed using the Benjamini–Hochberg procedure with a corrected false discovery rate (FDR) cut-off of 0.05. To find genes associated with drug sensitivity, too many genes were identified at the FDR cut-off of 0.05, so in order to find the most important drug response associated genes, we applied a more stringent cut-off of FDR<0.001.

Spearman rank correlation was used for assessing correlation between gene expression levels. Comparisons made between the standard method and our GRACE method are based on data from the same samples, i.e., samples without an available copy number will also be excluded from the standard analysis, whereas we include all available samples for the standard method in the GRACE web database.

Statistical analysis and data visualization were carried out using R.

**Correlation with orthogonal data sets**. In the CCLE data, we did not notice any saturated copy numbers. However, under-detection of gene expression was more prevalent, so we filtered out under-expressed genes with 0 values in over 50% of the total samples. For NCI-60 data, we filtered out genes with missing values in copy number data or gene expression data.

Spearman rank correlation was used for assessing correlation between protein levels, metabolite levels and gene expression levels, whereas Pearson correlation was used for correlation between drug sensitivity and gene expression levels[31].

**Implementation of the GRACE web database**. From the GRACE web database, users are prompted to input the gene name, choose the type of analysis, pick a TCGA cohort and select their preferred sample and method in a stepwise manner. Based on the input from each step, the options without available data will be disabled from the succeeding steps. The design of the analysis configuration and relational database follows the flow chart shown in Supplementary Fig. 5. The web interface was implemented in Javascript.

**Data availability**. All the data used in this study are from publically available data sets:

For TCGA data sets, we downloaded the normalized RNA-seq data processed by RSEM (RNA-Seq by Expectation Maximization) method and copy number data processed by GISTIC2 from GDAC Firehose (http://firebrowse.org/) from analysis run "02 April 2015"[32]. For CCLE data sets, we downloaded gene-centric RMA-normalized mRNA microarray expression data, DNA copy number data and pharmacological profiling data from the CCLE website (http://www.broadinstitute.org/ccle)[4]. For NCI-60 data sets, we downloaded the five-platform gene transcript-processed RNA data set and combined aCGH-processed DNA copy number data from the CellMiner database (http://discover.nci.nih.gov/cellminer/loadDownload.do)[33, 34]; we also downloaded the reverse protein lysate data set and Metabolon metabolomics data set from the NCI DTP Molecular Target program (https://wiki.nci.nih.gov/display/NCIDTPdata/Molecular+Target+Data).

The webtool we developed in this study allowing users to perform gene–gene co-expression study using TCGA data can be accessed through: https://grace.biohpc.swmed.edu/

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

## Acknowledgements

The authors would like to thank the BioHPC team in UT Southwestern Medical Center for help with web deployment and providing computational resources. We also thank Fang Huang from the DeBerardinis lab and Xiaowei Zhan from the QBRC for constructive suggestions for the GRACE web database, Jessie Norris for proofreading the manuscript, Xiaowei Zhan and Tao Wang for critical reading of the manuscript. This study was supported by the American Association for Cancer Research (AACR) Basic Cancer Research Fellowship (15-40-01-CAIL) awarded to L.C.; and grants from the National Cancer Institute (CA220449 to R.J.D., CA172211-01 to G.X., P50CA70907 to Y. X. and L.C.) and Cancer Prevention Research Institute of Texas (RP130272 to R.J.D., RP120732 to Y.X.).

## Author contributions

G.X.,R.J.D., and Y.X. supervised the project. L.C. and G.X. conceived the method. L.C. designed and performed the analyses, interpreted the results and developed the web application with advice from G.X. and Y.X. Y.D. deployed the web application. J.Y., Q.L., and G.X. wrote the mathematical expression of the method. L.C. drafted the article. G.X., Y.X., and R.J.D. critically edited the article.

## Additional information

**Competing interests:** The authors declare no competing financial interests.

