## [Peer Review File · Nature Communications]

Reviewers' comments:

Reviewer #1 (Remarks to the Author):

Cai et al. present a regression method for correction of RNA expression profiles in cancer samples. Somatic copy number alterations (SCNA) have previously been shown to alter RNA expression in a dose-dependent manner. Cai et al. correct for the copy-number dependent co-expression of genes, allowing true co-expression relationships to be seen.

The major contribution of this work is the development of the GRACE website, where users can browse co-expression relationships across TCGA cancer types. The portal was easy to use and informative, and it was easy to download data images and .csv files for further analysis or publication.

However, the analysis itself is not novel by any means. While the authors do cite some of the early works that detected a relationship between copy number and expression in human cancer, the authors should discuss in more detail recent genome-wide analyses such as Fehrmann et al., Nature 2015 and Ben-David et al., Nature Communications 2014.

Specific comments:

- Did the authors take tumor purity into account in any way? If not, the authors should add some discussion of how tumor purity might confound expression analysis and may or may not account for the differences observed between tumor and normal analyses
- Regarding analyses of "normal" samples from TCGA – many TCGA studies use peripheral blood as the normal sample. Were these removed from the analysis, or are the normal samples presented tissue-specific normals? If peripheral blood normal were used, then tissue-specific differences may underlie the differences noted in Figure 3b,c,e,f and the analysis and interpretation should be amended significantly.
- P.5 line 170, Authors state "As expected CCT4 co-expresses with genes encoding other subunits of the CCT complex in tumor, and this is better detected by GRACE than by the standard method (Figures 3d-e)" – However, the differences in panel E do not seem to be major. By what criteria are the authors making this statement?
- Minor comment – it is difficult to see the difference between the red color and purple/magenta color in Supplementary Figure 2.

Reviewer #2 (Remarks to the Author):

The paper presents a linear regression method to adjust the influence of copy number alterations on gene expression.

Somatic copy number alterations are frequent events in many cancer types such as breast cancer and high-grade serous ovarian cancer.

Copy number alterations (especially amplifications and homozygous deletions) typically correlate with mRNA expression.

Therefore, removing the effects of copy number alteration on expression is generally considered to be beneficial for many applications, e.g., investigating the transcription regulations in cancer,

The paper is well written and easy to follow.

I have some major concerns with the method itself, the results and how the results are presented and validated, which need to be addressed.

Major comments

Methods:

a) Previous studies have considered removing the influence of somatic copy number alterations on gene expression. For example, Aure, M.R., et al (PLOS One, 2013) also used linear regression to model copy number alterations on gene expression. In the xseq study (Ding, J. et al, Nature Communications, 2015), the authors used Gaussian process regression to model copy number alterations on gene expression.

b) It is not correct to say that the regression coefficient β is the same after introducing a new variable y (Line 294 - 295). In fact, the coefficient β will almost surely change after introducing a new feature.

c) The authors only consider pairwise gene expression influence (page 8). The authors should discuss the influences of the expression of multiple genes on the expression of a gene.

Results:

a) When presenting results, instead of just individual examples, the authors should systematically analyze the data and report all the results to support the findings because selectively looking at some examples may bias the conclusions.

For examples, when reporting the tumor and normal tissue-specific gene co-expression networks, the authors only gave the results for some example genes (PARP2 and CCT4). Instead, the author should systematically analyze and report all the genes with significant different co-expression networks between tumor and normal samples.

Similarly for reporting the results from analyzing the NCI-60 dataset and the CCLE dataset,

b) There are no validations for the presented results. For breast cancer, the METABRIC dataset (Curtis C. et al, Nature) with matched copy number and gene expression data for 2000 patients is a natural choice for validation purpose.

c) Several parts belong to 'Discussion' were presented in the 'Results' section (e.g., the sentence from line 163 to line 165)

Minors:

a) There are several arbitrary choose parameters without any justifications, e.g., an adjusted p-value thresholds of 0.001 was used in line 205, but an FDR cut-off of 0.05 was used in line 320. When selecting under-expressed genes, a threshold of 10% is used (line 314), a threshold of 5% is used for removing saturated copy number values (line 317), and a threshold of 50% is used to remove under-expressed genes (line 330).

b) Random variables are represented by capitalized letters (page 8, line 280)

c) mean zero and unit variance - zero mean and unit variance (page 8, line 282)

d) Consider - Considering (page 8, line 282)

We appreciate the thoughtful and generally positive comments from the reviewers, as well as their insights and helpful suggestions for improving the manuscript. We have carefully considered all comments and suggestions, and have addressed them in detail as outlined in the letter below. We hope that with these changes, the reviewers will find the manuscript acceptable for publication.

The reviewers' comments verbatim are in blue italics. Our responses are in plain black text.

REVIEWER #1 (REMARKS TO THE AUTHOR):

“Cai et al. present a regression method for correction of RNA expression profiles in cancer samples. Somatic copy number alterations (SCNA) have previously been shown to alter RNA expression in a dose-dependent manner. Cai et al. correct for the copy-number dependent co-expression of genes, allowing true co-expression relationships to be seen.

The major contribution of this work is the development of the GRACE website, where users can browse co-expression relationships across TCGA cancer types. The portal was easy to use and informative, and it was easy to download data images and .csv files for further analysis or publication.”

We thank the reviewer for these positive comments on our website! We are also grateful to the reviewer for the positive comments about our manuscript and for recognizing the potential impact of our work. We have responded to the reviewer's comments as follows.

Major suggestions for improvement:

• Comment 1

“However, the analysis itself is not novel by any means. While the authors do cite some of the early works that detected a relationship between copy number and expression in human cancer, the authors should discuss in more detail recent genome-wide analyses such as Fehrmann et al., Nature 2015 and Ben-David et al., Nature Communications 2014. “

Response:

We thank the reviewer for pointing us to these important papers, and we have cited these papers in our revisions. It is important to notice that Fehrmann et al., Nature 2015; Ben-David et al., Nature Communications 2014 and several papers (Pollack, Sorlie et al. 2002, Tsafrir, Bacolod et al. 2006, Zack, Schumacher et al. 2013) reported the correlations between CNV and expression data. However, no method exists to remove the confounding effect of copy number alterations in the analysis of gene-gene co-expression. This is the first study to show strong evidence that CNVs bias co-expression studies. Thus, the GRACE approach is the first to correct this bias with the goal of identifying bona fide co-regulated genes.

• **Comment 2**

“the authors take tumor purity into account in any way? If not, the authors should add some discussion of how tumor purity might confound expression analysis and may or may not account for the differences observed between tumor and normal analyses”

Response:

We thank the reviewer for pointing out this important issue. We have performed the analysis and added this to our Discussion and expanded Supplementary Figure 3 (also included below in the response letter).

To test the effect of mixing normal cells and tumor cells, we have generated synthetic data by mixing expression values from 110 pairs of matched tumor and adjacent normal samples at different ratios and comparing the distribution of correlation. We observed a gradual increase in tail distribution (i.e. an increasing proportion of high correlation coefficients) as the proportion of normal sample expression values increases (Supplementary Figure 3d). Therefore, the gene-gene correlation is generally higher in normal samples than in tumor samples. TCGA has implemented quality control procedures to ensure each tumor sample has at least 60% tumor nuclei. But we still do not know the extent of impact from tumor impurity in the co-expression analysis. The design of TCGA does not allow us to address such problems, which is a limitation of using bulk tumor for omics analysis. However, because the majority of tumor genomic data generated so far are still from bulk tumors, our method will be important for the analysis of such data. In the future, with the accumulation of large-scale single cell sequencing data, or with the advent of mature methods to computationally determine expression profiles of tumor, stromal and immune expression from bulk tumor expression measurement, we would love to test our method on the new datasets and we would expect it to work better with purer data.

Supplementary Figure 3. Limitations of GRACE

a, Copy number versus RNA levels of ERBB2 from tumor samples. **b**, Kernel density estimation plots that visualize the distribution of pooled Spearman rank correlation coefficients for pairwise correlation from all the genes using tumor samples (by standard method or GRACE) or normal samples. Analyses are based on TCGA BRCA data. **c**, The number of significant pairwise gene correlations calculated from normal tissue data is higher than that from the tumor tissue data. **d**, Distribution of correlation coefficients from synthetic samples that had matched tumor and normal sample expression data mixed together at different ratios.

• **Comment 3**

“- Regarding analyses of “normal” samples from TCGA – many TCGA studies use peripheral blood as the normal sample. Were these removed from the analysis, or are the normal samples presented tissue-specific normals? If peripheral blood normal were used, then tissue-specific differences may underlie the differences noted in Figure 3b,c,e,f and the analysis and interpretation should be amended significantly.”

Response:

In our analysis, we did not include any peripheral blood samples. Specifically, the sample type information for TCGA samples was encoded in the same ID according to <https://wiki.nci.nih.gov/display/TCGA/TCGA+barcode>. We have checked all the normal samples from the 10 TCGA cohorts we used in our analyses and confirmed that they all have sample code 11, which means they are all “Solid Tissue Normal (NT)”.

• **Comment 4**

“- P.5 line 170, Authors state “As expected CCT4 co-expresses with genes encoding other subunits of the CCT complex in tumor, and this is better detected by GRACE than by the standard method (Figures 3d-e)” – However, the differences in panel E do not seem to be major. By what criteria are the authors making this statement?”

Response:

In the original figure 3e, we compared the results for the top 100 genes. As the reviewer pointed out, the advantage of using GRACE to examine co-expressing genes of CCT4 is less obvious than the case for PARP2 in Figure 3b. This is because the gene set for which CCT4 co-expressing genes are enriched in (“FORMATION_OF_TUBULIN_FOLDING_INTERMEDIATES_BY_CCT_TRIC” from Reactome) contains much fewer genes than the gene set for which PARP2 co-expressing genes are enriched in (“CELL_CYCLE” from Reactome); it is 22 genes versus 421 genes. Although genes from “FORMATION_OF_TUBULIN_FOLDING_INTERMEDIATES_BY_CCT_TRIC” have lower ranks in the CCT4 co-expressing genes detected by the standard method compared to the GRACE method, they are still within the top 100. For this reason, we have remade Figure 3e with the top 20 CCT4 co-expressing genes.

Previous Fig.3e

Current Fig.3e

Figure 3. Examples of co-expressing genes differing in tumor vs normal tissues

e, Frequencies of genes from “REACTOME_FORMATION_OF_TUBULIN_FOLDING_INTERMEDIATES_BY_CCT_TRIC” in top 20 CCT4 co-expressing genes. Analyses are based on TCGA BRCA data. P-values from hypergeometric tests are given next to the bars.

• **Comment 5**

“Minor comment – it is difficult to see the difference between the red color and purple/magenta color in Supplementary Figure 2”

Response: We have changed the color for normal from magenta to green in both Figure 3 and Supplementary Figure 2.

REVIEWER 2 (REMARKS TO THE AUTHOR):

“The paper presents a linear regression method to adjust the influence of copy number alterations on gene expression.

Somatic copy number alterations are frequent events in many cancer types such as breast cancer and high-grade serous ovarian cancer.

Copy number alterations (especially amplifications and homozygous deletions) typically correlate with mRNA expression.

Therefore, removing the effects of copy number alteration on expression is generally considered to be beneficial for many applications, e.g., investigating the transcription regulations in cancer.

The paper is well written and easy to follow.”

We thank the reviewer for the positive feedback and detailed comments on how to improve the manuscript. We have responded to the reviewer’s comments as follows.

Major comments:

• Methods:

“a) Previous studies have considered removing the influence of somatic copy number alterations on gene expression. For example, Aure, M.R., et al (PLOS One, 2013) also used linear regression to model copy number alterations on gene expression. In the xseq study (Ding, J. et al, Nature Communications, 2015), the authors used Gaussian process regression to model copy number alterations on gene expression.”

Response:

We thank the reviewer for pointing us to these important papers, and we have now cited these papers in the introduction. However, we want to emphasize that the main goal of our GRACE method is different from previous papers. The previous papers emphasized the causal relationship between genomic aberrations (copy number alterations in the paper by Aure, M.R., et al; somatic mutation in the paper by Ding, J. et al) and transcription, whereas our paper is the first to apply copy number adjusted gene expression in addressing the question of gene-gene co-expression. We have shown strong evidence that the CNV will bias the co-expression analysis, and we have also provided a new method to correct such bias and identify real co-expressing genes. In addition, we provide a user-friendly website that allows the general public to examine the co-expressing genes for their favorite genes by the traditional method and the copy number adjusted method, in cancer samples as well as in normal samples.

b) It is not correct to say that the regression coefficient beta is the same after introducing a new variable y (Line 294 - 295). In fact, the coefficient beta will almost surely change after introducing a new feature.

Response:

We thank the reviewer for pointing this out. We have deleted the corresponding sentence.

c) The authors only consider pairwise gene expression influence (page 8). The authors should discuss the influences of the expression of multiple genes on the expression of a gene.

Response:

This is an important point. In the revision, we have conducted a new simulation to discuss such circumstance. Specifically, we generated a batch of synthetic gene expression datasets from Gaussian Graphical Models that consider partial correlations (the influences of expression of multiple gens) and incorporate copy number effects. We found that the estimated pairwise correlations obtained by GRACE are very close to the true partial correlations, while the estimated pairwise correlation without using GRACE method performs worse. We have included these new results in the supplement and have also copied the main results here (Please see the Supplementary section for details):

“To assess the accuracy of GRACE under different noise levels, $\varepsilon = 0, 0.1, 0.5, 1$ and different network models, including autoregressive (AR) model, Barabási-Albert (BA) model, and Erdős-Rényi (ER) model, for the influence of other genes, we generated 12 groups of synthetic datasets. For each group, 100 datasets were independently simulated. To quantify the performance, we used the root-mean-square error (RMSE) to measure the differences between the true correlation matrix $\mathbf{P} = (\text{diag}(\boldsymbol{\Sigma}))^{-1/2} \boldsymbol{\Sigma} (\text{diag}(\boldsymbol{\Sigma}))^{-1/2}$ and the estimated one $\hat{\mathbf{P}}$ by GRACE, calculated by $RMSE = \frac{\sum_{j < h} (\rho_{jh} - \hat{\rho}_{jh})^2}{p(p-1)/2}$. The boxplot of RMSEs under different settings are displayed in Supplementary Notes Figure N1. It shows that the estimated correlations by GRACE are good approximations of their true values, especially when the noise level is at a low level. Among the three network models, there is not much difference when $\varepsilon < 0.5$. However, if the noise level becomes stronger, the AR and BA models outperform the ER model. We also plot the scatter points of the true and estimated correlation matrix for one of the synthetic datasets in the group, for which $\varepsilon = 0.1$ and the network model is ER. As shown in Supplementary Notes Figure N2, again, the estimated correlations by GRACE are good approximations of the truth with $RMSE = 0.00093$. In summary, GRACE provides a good approximation of the partial correlation, and it greatly improves the computation efficiency. Furthermore, Supplementary Notes Figure N3 show the result when we directly calculated the correlations of expression levels between each pair of genes, without considering the effect of copy number values on gene expression levels. As we can see, it fails to recover the truth and results in a number of false positives, so it is important to adjust the copy number in the co-expression analysis.”

Supplementary Notes Figure N1. The boxplots of RMSEs under different network models (AR, BA, and ER) and different noise levels ϵ

Supplementary Notes Figure N2. The scatter plot of the upper-triangle entries of the true correlation matrix $(\text{diag}(\Sigma))^{-1/2} \Sigma (\text{diag}(\Sigma))^{-1/2}$ and of the estimated correlation matrix by GRACE.

Supplementary Notes Figure N3. The scatter plot of the upper-triangle entries of the true correlation matrix $(\text{diag}(\Sigma))^{-1/2}\Sigma(\text{diag}(\Sigma))^{-1/2}$ and of the correlation matrix of gene expression levels data $\text{corr}(Y_1 \ \dots \ Y_p)$ without using GRACE.

• **Results:**

a) When presenting results, instead of just individual examples, the authors should systematically analyze the data and report all the results to support the findings because selectively looking at some examples may bias the conclusions.

Response:

Following the reviewer’s suggestion, we have now included results from our systematic analysis in the supplementary tables. We have included detailed responses below:

For examples, when reporting the tumor and normal tissue-specific gene co-expression networks, the authors only gave the results for some example genes (PARP2 and CCT4). Instead, the author should systematically analyze and report all the genes with significant different co-expression networks between tumor and normal samples.

Response:

We have generated results for 10 TCGA cohorts that have both normal and tumor samples. For each cohort, each gene and each gene-set from a given gene-set library, we determined whether the enrichment of the top 100 co-expressing genes in the gene-set is only significant in tumor or normal samples. We include the top 500 results for each cohort in supplementary tables S1-S4. The snapshot below is an example from analysis using TCGA BRCA cohort expression data and gene family classification from HUGO Gene Nomenclature Committee

(HNGC). These top 4 genes encode protocadherins, and in tumor samples they are all highly correlated with other protocadherin genes, whereas in the normal tissues, the co-expression is insignificant.

1	BRCA			
2	genes	set.names	tumor	normal
3	PCDHB8	Clustered_protocadherins	4.90E-89	0.2717164
4	PCDHB2	Clustered_protocadherins	6.66E-86	0.2717164
5	PCDHGA7	Clustered_protocadherins	7.74E-80	1
6	PCDHGA8	Clustered_protocadherins	7.74E-80	1

Figure R2.1 Snapshot of supplementary Table S1

Similarly for reporting the results from analyzing the NCI-60 dataset and the CCLE dataset,
Response:

We have generated supplementary files to include results from the systematic analyses for NCI-60 and CCLE dataset.

Supplementary table S5 records the number of genes that positively correlate with metabolites with q-value < 0.05 (snapshot in figure R2.2). By one-sided Wilcoxon signed rank test, the number of significant genes called by RES is significantly more than by RNA (p-value 3.825e-09).

Supplementary table S6 records the number of genes that positively correlate with RPPA with q-value < 0.05. By one-sided Wilcoxon signed rank test, the number of significant genes called by RES is significantly more than by RNA (p-value 7.223e-04).

Supplementary table S7 records the number of genes that negatively correlate with maximal effect level of each drug with q-value < 0.001. These are the same values used to plot Figure 4d.

1		RNA	RES
2	(2-Aminoethyl)phosphonate	2	4
3	(p-Hydroxyphenyl)lactic_acid	0	0
4	2,3-diphospho-D-glyceric_acid	0	0
5	2'-deoxyadenosine_5'-diphosphate	1	1
6	2'-deoxyuridine_5'-triphosphate	0	0
7	2'-deoxyuridine	22	13
8	3-hydroxy-3-methylglutarate	146	387
9	3-methyl-L-histidine	5	7
10	3-phospho-d-glycerate	142	263

Figure R2.2 Snapshot of supplementary table S5

b) There are no validations for the presented results. For breast cancer, the METABRIC dataset (Curtis C. et al, Nature) with matched copy number and gene expression data for 2000 patients is a natural choice for validation purpose.

Response:

In our manuscript, we showed with a systematic analysis that GRACE can reduce intra-chromosomal gene-gene correlation while increasing inter-chromosomal gene-gene correlation in Figure 2 using TCGA data. In our original paper, we validated this with CCLE data in Supplementary Figure 1. Following the reviewer’s suggestion, we performed another validation analysis with the METABRIC data and have added the results next to the CCLE validation in Supplementary Figure 1. From this analysis, we again validated that GRACE can reduce intra-chromosomal gene-gene correlation while increasing inter-chromosomal gene-gene correlation.

Figure R2.3 Same as the new supplementary figure S1 (Left panels are results from CCLL data analysis, and the right panels are from METABRIC data analysis)

c) Several parts belong to 'Discussion' were presented in the 'Results' section (e.g., the sentence from line 163 to line 165)

Response:

We have made changes accordingly. Specifically, we have removed the sentence from lines 163-165 from the original paper (“It is possible that while the inhibition of PARP1 by PARP inhibitors could be beneficial for patients, the simultaneous inhibition of PARP2 in normal tissues could lead to more side effects by dysregulating transcription.”) and added the following sentences to the end of the first paragraph in the Discussion section:

“Our method can also help to discern the differences in transcriptional co-regulation between tumor and normal tissues. For example, we showed that whereas both PARP1 and PARP2 co-express with cell cycle genes in tumor samples consistent with their known roles in DNA repair, only PARP2 strongly co-expresses with many C2H2 type zinc finger genes in normal tissues. It is possible that while the inhibition of PARP1 by PARP inhibitors could be beneficial for patients, the simultaneous inhibition of PARP2 in normal tissues could lead to more side effects by dysregulating transcription.”

• **Minors:**

a) There are several arbitrary choose parameters without any justifications, e.g., an adjusted p-value thresholds of 0.001 was used in line 205, but an FDR cut-off of 0.05 was used in line 320. When selecting under-expressed genes, a threshold of 10% is used (line 314), a threshold of 5% is used for removing saturated copy number values (line 317), and a threshold of 50% is used to remove under-expressed genes (line 330).

Response:

In this revision, we have added justifications for different parameters.

We used different FDR cut-off values for different types of analysis based on the variability and quality of the data. For example, when we use the standard FDR 5% as a cut-off to select drug response genes (original line 205), we identified too many genes. Although our proposed method is still better than the standard method (Figure below), this FDR cutoff could not lead to the most important drug response gene set discovery. Therefore, we used a more stringent cutoff (FDR<0.1%) to identify the most important drug response gene sets.

For different datasets, the noise level is different so we need to apply different selection cut-offs to make sure that we can still retain a good proportion of the data while controlling for its quality. For example, as we have already noted in the original manuscript, TCGA mRNA expression data was generated by RNA-seq while CCLE was generated by microarray, so TCGA has much less missing data compared to the CCLE data. If we use the same data selection cutoff for two datasets, too many genes will be removed from the CCLE data. Therefore, a threshold of 10% is used (original manuscript, line 314) for TCGA data and a threshold of 50% is used (original manuscript, line 330) for CCLE data.

Figure R2.4 More genes associated with drug sensitivity by residuals than by RNA at $q < 0.05$ cutoff

b) Random variables are represented by capitalized letters (page 8, line 280)

Response:

We thank the reviewer for pointing out this typo. We have capitalized all the random variables in the Materials and Methods section.

c) mean zero and unit variance - zero mean and unit variance (page 8, line 282)

Response:

We thank the reviewer for pointing out this typo. We have corrected it. Please see Line 303, Page 8.

d) Consider - Considering (page 8, line 282)

Response:

We thank the reviewer for pointing out this typo. We have corrected it. Please see Line 303, Page 8.

Pollack, J. R., T. Sorlie, C. M. Perou, C. A. Rees, S. S. Jeffrey, P. E. Lonning, R. Tibshirani, D. Botstein, A. L. Borresen-Dale and P. O. Brown (2002). "Microarray analysis reveals a major direct role of DNA copy number alteration in the transcriptional program of human breast tumors." Proc Natl Acad Sci U S A **99**(20): 12963-12968.

Tsafrir, D., M. Bacolod, Z. Selvanayagam, I. Tsafrir, J. Shia, Z. Zeng, H. Liu, C. Krier, R. F. Stengel, F. Barany, W. L. Gerald, P. B. Paty, E. Domany and D. A. Notterman (2006). "Relationship of gene expression and chromosomal abnormalities in colorectal cancer." Cancer Res **66**(4): 2129-2137.

Zack, T. I., S. E. Schumacher, S. L. Carter, A. D. Cherniack, G. Saksena, B. Tabak, M. S. Lawrence, C. Z. Zhsng, J. Wala, C. H. Mermel, C. Sougnez, S. B. Gabriel, B. Hernandez, H. Shen, P. W. Laird, G. Getz, M. Meyerson and R. Beroukhim (2013). "Pan-cancer patterns of somatic copy number alteration." Nat Genet **45**(10): 1134-1140.

Reviewers' comments:

Reviewer #1 (Remarks to the Author):

The authors have sufficiently addressed my questions and improved the manuscript.

One typographical error was noticed - Figure 1 - panel e and f are in the wrong order (panel e should be on the left and panel f on the right - and/or consult the editor for recommendations).

Reviewer #2 (Remarks to the Author):

The first round of reviews was detailed and the authors responded well to my comments.

Some minor points:

a, for validation, it's good that the authors provided the METABRIC data results. However, my major concern is the conservations of the co-expression networks, e.g., are the co-expression networks from the TCGA breast cancer data and the METABRIC breast cancer datasets well conserved after correction?

b, for copy number data, because GISTIC truncates very large copy number values, e.g., ERBB2 amplifications, the authors could consider using the original copy number log₂ values instead for their analyses.

c, the order of the sub-figures, especially in Figure 1 is a little bit messy, considering reordering them.

We appreciate the positive comments from the reviewers, as well as their insights and helpful suggestions for improving the manuscript. We have carefully considered all comments and suggestions, and have addressed them in detail as outlined in the letter below. We hope that with these changes, the reviewers will find the manuscript acceptable for publication.

The reviewers' comments verbatim are in blue italics. Our responses are in plain black text.

REVIEWER #1 (REMARKS TO THE AUTHOR):

“The authors have sufficiently addressed my questions and improved the manuscript..”

We are very pleased to hear that the reviewer is satisfied with our revision.

Comments:

• Comment 1

“One typographical error was noticed - Figure 1 - panel e and f are in the wrong order (panel e should be on the left and panel f on the right - and/or consult the editor for recommendations. “

Response:

We thank the reviewer for this suggestion. In fact, this was actually not a typographical error. We originally tried to align Figure 1c-e together on the right panel because these three figures are all based on analyses of chromosome 1, and we wanted to allow readers to focus on the patterns of p and q arms in the three figures. We wanted to show that copy number variation (Figure 1c) leads to the correlation between gene expression and copy number (Figure 1d) and further results in the increased positive gene expression correlation among neighboring genes (Figure 1e). However, since both reviewers found this ordering confusing, we have re-ordered the figures in the revised manuscript (Figure R1.1).

Figure R1.1 Reordered Figure 1 for manuscript

REVIEWER 2 (REMARKS TO THE AUTHOR):

“The first round of reviews was detailed and the authors responded well to my comments..”

We are very pleased to hear that the reviewer is satisfied with our revision.

Comments:**• Comment 1**

“a, for validation, it's good that the authors provided the METABRIC data results. However, my major concern is the conservations of the co-expression networks, e.g., are the co-expression networks from the TCGA breast cancer data and the METABRIC breast cancer datasets well conserved after correction?”

Response:

We thank the reviewer for clarifying their concerns. Ideally, the co-expression networks from the same cancer type would be consistent across different datasets, and whether the co-expression networks are conserved or not can be a good criterion to evaluate the computation methods. However, empirical studies show that the co-expression networks are often inconsistent across datasets for the following reasons: 1. The inherent data noise exists in different degrees in the genome-wide mRNA expression data. 2. Different profiling platforms across studies may lead to some systematic differences among correlation patterns. 3. Differences in tumor purity - as shown in our Supplementary Fig 3d, the co-expression patterns are quite different among tumor samples and normal samples, and this difference may lead to differences in co-expression patterns. 4. Heterogeneity of tumor samples - studies have shown that intra-tumor heterogeneity leads to different mRNA expression profiles. 5. Different clinical characteristics/sub-populations from different cohorts. When comparing the METABRIC breast cancer and TCGA breast cancer cohorts, 48% of the METABRIC cohort has stage III breast cancer, while only 28% of patients are stage III from the TCGA cohort. 12% of the METABRIC patients are HER2-enriched subtype, while only 6.3% are HER2-enriched in the TCGA cohort. In addition, mRNA expression was measured using the Illumina HT-12 platform in METABRIC, while TCGA gene expression profile was measured using the Illumina HiSeq 2000 RNA Sequencing platform. Due to this cohort and platform difference between METABRIC and TCGA data, there will inevitably be some inherent differences in co-expression networks between the two datasets. Therefore the consistency between these two datasets may not be a good criterion for evaluating the GRACE method vs. standard method.

Nevertheless, we compared the consistency of co-expression networks between METABRIC and TCGA by using the GRACE and standard methods. To assess the consistency of co-expression networks, we used the correlation of correlations to measure the homogeneity of

the co-expression structure generated by unadjusted RNA expression (RNA) or residuals from regressing RNA on copy number (RES) using TCGA-BRCA or Metabric data. This metric is built upon the concept of integrative correlation (IGC) (Garrett-Mayer, Parmigiani et al. 2008) and can be used to evaluate the reproducibility across studies (Kang, Sibille et al. 2012). We found the correlation between correlation matrices generated by the standard method (RNA) and the GRACE method (RES) are highly correlated from the same cohort (BRCA.RNA vs BRCA.RES: 0.93, Metabric.RNA vs Metabric.RES: 0.95). Positive correlation could also be observed between the two different cohorts. The cross-cohort correlation is higher between correlation matrices generated by the standard method (BRCA.RNA vs Metabric.RNA: 0.50) than by the GRACE method (BRCA.RES vs Metabric.RES: 0.45), suggesting that the co-expression structure is more reproducible between cohorts with the standard method compared to the GRACE method (Figure R2.1).

Figure R2.1 Correlation among correlation matrices constructed based on the standard method or GRACE for Metabric or TCGA-BRCA data

However, both biological transcriptional regulation and copy number variation introduced bias affect the co-expression network. The higher similarity captured by the standard approach than our GRACE method could be a result of the high reproducibility of copy number-originated noise in both breast cancer cohorts. We hypothesized that since different types of cancer have different copy number variation patterns, the CNV impact on the co-expression network would also differ, and hence the cross-cancer-cohort correlation based on the standard method would be less than that based on the GRACE method. Two steps were taken to test this hypothesis.

As the first step, we tested if there is a higher degree of copy number variation similarity between the BRCA and Metabric data than between BRCA and other non-breast cancer TCGA cohorts. To test this hypothesis, we calculated the correlation of copy number correlation matrices (Figure R2.2). Indeed, a high degree of similarity was observed between BRCA and Metabric, and this implies that the impact of copy number variation-originated

systematic noise on mRNA co-expression network would be more similar between BRCA and Metabric cohort than between BRCA and other non-breast cancer TCGA cohorts.

Correlation of copy number correlation matrix between TCGA-BRCA and selected cohorts

Figure R2.2 Similarity of copy number variation between TCGA-BRCA cohort and selected cohorts

In the second step, to test whether lower similarity in copy number variation between BRCA and other non-breast cancer TCGA cohorts could lower the converseness in co-expression networks captured by the standard method in comparison to the GRACE method, we applied the same analysis narrated for Figure 2.1 (for BRCA vs Metabric comparison) to compare BRCA with 18 other TCGA cancer cohorts. In all 18 cases, we indeed observed higher cross-cancer-cohort correlation using GRACE (Figure R2.3).

Figure R2.2 Correlation among correlation matrices constructed based on standard method or GRACE for Metabric or TCGA-BRCA data

From the above findings, we show that co-expression network in cancer is influenced by copy number variation, and we believe this would confound the interpretation of co-expression network in terms of functional relevance.

We analyzed the KEGG_RIBOSOME gene set as an example. When we look at the distribution of all pairwise correlation coefficient for genes within this gene set, in both cohorts, the vast majority of the correlation coefficients are positive. Importantly, GRACE produced more positive correlation than the standard method (Figure R2.4, median values are marked by vertical lines), suggesting that GRACE has better performance in capturing a biologically meaningful co-expression network.

Figure R2.4 Distribution of all pairwise correlation among KEGG ribosomal genes by Standard method or GRACE in BRCA or Metabric cohorts

Besides the above example, we have systematically evaluated over 1000 canonical pathway gene sets downloaded from the Molecular Signatures Database (<http://software.broadinstitute.org/gsea/msigdb>), and we provide in Table R2.1 (snapshot shown below in Figure R2.5) the median value of pairwise correlation coefficients for genes within each specific gene set from the four settings.

	BRCA.RNA	BRCA.RES	Metabric.RN	Metabric.RES
KEGG_GLYCOLYSIS_GLUONEOGENESIS	0.01498116	0.03611167	0.01012482	0.02208113
KEGG_CITRATE_CYCLE_TCA_CYCLE	0.09873634	0.17526138	0.07555515	0.13091781
KEGG_PENTOSE_PHOSPHATE_PATHWAY	0.03048966	0.06439457	0.02728461	0.04350736
KEGG_PENTOSE_AND_GLUCURONATE_INTERCONVERSIONS	-0.0260951	0.02510802	-0.0027385	0.00358781
KEGG_FRUCTOSE_AND_MANNOSE_METABOLISM	0.03222328	0.06143273	0.02549598	0.03992052
KEGG_GALACTOSE_METABOLISM	0.02532369	0.04981145	0.00351815	0.01279435
KEGG_ASCORBATE_AND_ALDARATE_METABOLISM	0.01092524	0.00712972	-0.0003939	-0.0007256
KEGG_FATTY_ACID_METABOLISM	0.04020531	0.0378769	0.03707954	0.04877384
KEGG_STEROID_BIOSYNTHESIS	0.06655808	0.06822518	0.05351356	0.07862851
KEGG_PRIMARY_BILE_ACID_BIOSYNTHESIS	0.01380659	-0.0028607	0.0105005	0.00673362
KEGG_STEROID_HORMONE_BIOSYNTHESIS	0.03709976	0.03214261	0.00272709	0.00233407
KEGG_OXIDATIVE_PHOSPHORYLATION	0.19134019	0.27521152	0.07888925	0.11381052
KEGG_PURINE_METABOLISM	-0.0004444	0.01301647	0.0017504	0.01019537
KEGG_PYRIMIDINE_METABOLISM	0.0383282	0.0646679	0.02313095	0.04087391
KEGG_ALANINE_ASPARTATE_AND_Glutamate_METABOLISM	-0.0040153	-0.0025313	-0.0039427	0.00535761
KEGG_GLYCINE_SERINE_AND_THREONINE_METABOLISM	0.01362112	0.01985854	0.00320233	0.00755576

Figure R2.5

In most cases like KEGG_RIBOSOME, a higher median pairwise correlation coefficient is found with our GRACE method compared to the standard method (Figure R2.6)

median pairwise correlation coefficient within gene sets

Figure R2.6

In summary, due to the inherent differences between METABRIC and TCGA data, inherent differences of co-expression network exist between two datasets, so the consistency between these two datasets may not be a good criterion for evaluating the GRACE method vs. standard method. When we used the correlation of correlations method to compare the correlation generated from the standard method and GRACE method, the two methods generated very similar expression networks within the same cohort. The expression networks in BRCA and Metabric generated by the standard method and GRACE are both conserved, but the standard method generates more consistent results between the two cohorts, which is due to the high similarity of copy number variation patterns in both breast cancer cohorts and hence similarity in the impact of copy number variation on the co-expression structure. Consistent with this notion, the extent of conservation for the co-expression structure of BRCA and 18 other TCGA

cancer cohorts are found to be greater by the GRACE method than by the standard method. We also found stronger co-expression of functionally associated genes defined by curated canonical pathways by using GRACE compared to the standard method, suggesting that GRACE has better performance in capturing biologically meaningful co-expression networks.

• **Comment 2**

“b, for copy number data, because GISTIC truncates very large copy number values, e.g., ERBB2 amplifications, the authors could consider using the original copy number log2 values instead for their analyses.”

Response:

GISTIC (Mermel, Schumacher et al. 2011) is a mature copy number analysis pipeline and has been used routinely in the analysis workflow for TCGA publications. We have therefore chosen to use copy number estimation from GISTIC for analysis in our manuscript.

Following the reviewer’s suggestion, we downloaded the segmented copy number data for the TCGA-BRCA cohort from <http://firebrowse.org/> and converted it to gene-level data. Segmentation is the process of taking noisy intensity measurements into chromosomal regions of equal copy number. The same segmented copy number data was used as input for the GISTIC pipeline that produced the data we used in our original analysis.

For the samples with ERBB2 relative copy number value truncated at 3.657 by GISTIC, the copy number derived from the segmented copy number data is still continuous (Figure R2.7a). However, the copy number derived from the segmented copy number data does not follow a linear relationship with the RNA expression data (Figure R2.7a). In contrast, for EIF2D (used as an example gene in Figure 2 of our paper), the copy number for both the GISTIC pipeline and segmented data, and RNA expression are all in a linear relationship to each other (Figure R2.7b) and GRACE was able to correct the copy number-based neighbor gene bias in EIF2D co-expressing genes using the segmented copy data (Figure R2.8).

Figure R2.7 Relationship among GISTIC output copy number, segmented copy number and RNA expression for ERBB2 and EIF2D

Figure 2.8 Removal of neighboring genes by GRACE using segmented copy number data

We therefore speculate that the problem with ERBB2 originated from the experimental measurement. For example, the signal was saturated because the input DNA material exceeded the linear range limit for ERBB2 copy number detection. Unfortunately, we do not have a good way to computationally correct this problem, so we still need to remove such genes for our analysis.

• **Comment 3**

“c, the order of the sub-figures, especially in Figure 1 is a little bit messy, considering reordering them.”

Response:

We thank the reviewer for this suggestion. In fact, we originally tried to align Figures 1c-e together on the right panel because these three figures are all based on analyses of chromosome 1, and we wanted to allow readers to focus on the patterns of p and q arms in the three figures. We wanted to show that copy number variation (Figure 1c) leads to the correlation between gene expression and copy number (Figure 1d) and further results in the increased positive gene expression correlation among neighboring genes (Figure 1e). However, since both reviewers found this ordering confusing, we have re-ordered the figures in the revised manuscript (Figure R1.1).

Reference:

- Garrett-Mayer, E., G. Parmigiani, X. Zhong, L. Cope and E. Gabrielson (2008). "Cross-study validation and combined analysis of gene expression microarray data." Biostatistics **9**(2): 333-354.
- Kang, D. D., E. Sibille, N. Kaminski and G. C. Tseng (2012). "MetaQC: objective quality control and inclusion/exclusion criteria for genomic meta-analysis." Nucleic Acids Res **40**(2): e15.
- Mermel, C. H., S. E. Schumacher, B. Hill, M. L. Meyerson, R. Beroukhim and G. Getz (2011). "GISTIC2.0 facilitates sensitive and confident localization of the targets of focal somatic copy-number alteration in human cancers." Genome Biol **12**(4): R41.

REVIEWERS' COMMENTS:

Reviewer #2 (Remarks to the Author):

The authors were very responsive to the reviews and addressed my concerns. As a result, the evidence for the performance of GRACE is highlighted.

Response to Reviewer

The reviewer' comment verbatim are in blue italics. Our responses are in plain black text.

REVIEWER #2 (REMARKS TO THE AUTHOR):

The authors were very responsive to the reviews and addressed my concerns. As a result, the evidence for the performance of GRACE is highlighted.

We appreciate the acceptance and recognition of our work by the reviewer.